DISCOVERY REPORT

# A functional bacteria-derived restriction modification system in the mitochondrion of a heterotrophic protist

**David S. Milner**[1], **Jeremy G. Wideman**[2,3]*, **Courtney W. Stairs**[4], **Cory D. Dunn**[5], **Thomas A. Richards**[1]*

**1** Department of Zoology, University of Oxford, Oxford, United Kingdom, **2** Biodesign Center for Mechanisms of Evolution, School of Life Sciences, Arizona State University, Tempe, Arizona, United States of America, **3** Wissenschaftskolleg zu Berlin, Berlin, Germany, **4** Department of Biology, Lund University, Lund, Sweden, **5** Institute of Biotechnology, Helsinki Institute of Life Science, University of Helsinki, Helsinki, Finland

☙ These authors contributed equally to this work.
* jeremy.wideman@asu.edu (JGW); thomas.richards@zoo.ox.ac.uk (TAR)

## Abstract

The overarching trend in mitochondrial genome evolution is functional streamlining coupled with gene loss. Therefore, gene acquisition by mitochondria is considered to be exceedingly rare. Selfish elements in the form of self-splicing introns occur in many organellar genomes, but the wider diversity of selfish elements, and how they persist in the DNA of organelles, has not been explored. In the mitochondrial genome of a marine heterotrophic katablepharid protist, we identify a functional type II restriction modification (RM) system originating from a horizontal gene transfer (HGT) event involving bacteria related to flavobacteria. This RM system consists of an HpaII-like endonuclease and a cognate cytosine methyltransferase (CM). We demonstrate that these proteins are functional by heterologous expression in both bacterial and eukaryotic cells. These results suggest that a mitochondrion-encoded RM system can function as a toxin–antitoxin selfish element, and that such elements could be co-opted by eukaryotic genomes to drive biased organellar inheritance.

## Introduction

Endosymbiosis, the localisation and functional integration of one cell within another [1–3], can lead to the evolution of specialised organellar compartments responsible for a range of cellular and biochemical functions [4]. Mitochondria and plastids originate from endosymbiotic events and typically retain vestigial genomes of bacterial ancestry [5,6]. While sequencing initiatives have demonstrated that mitochondrial gene content can vary extensively, their evolution in every eukaryotic lineage is typified by both functional and genomic reduction [7,8]. Rare gene replacements and novel gene acquisitions into mitochondrial genomes have been identified [9,10], particularly involving plant-to-plant gene transfers [11–14], with plants also susceptible to the transfer of entire organelles, and their genomes, from one cell to another [15–17]. Moreover, mitochondrial group I and II self-splicing introns demonstrate a pattern

**Data Availability Statement:** The authors confirm that all data underlying the findings are fully available without restriction. The single-cell amplified genome assemblies and the K1, K3 and K4 mitochondrial genome contigs, originally from Wideman et al., 2020, are available at https://doi.org/10.6084/m9.figshare.7352966 and https://doi.org/10.6084/m9.figshare.7314728, respectively. Alignment data for Figs 2A and S1 are available at http://doi.org/10.6084/m9.figshare.c.5336963. Amino acid sequences for proteins Kat-HpaII, Kat-HpaII-CM, Kat-MutH, Kat-MutH-CM, Algibacter-HpaII (accession: WP_054724019) and Algibacter-HpaII-CM (accession: WP_054724021) are available in S3 Table. All additional relevant data are within the paper and its Supporting Information files.

**Funding:** C.D.D. is supported by the Sigrid Jusélius Foundation, the Academy of Finland (https://www.aka.fi, grant no. 331556), and the Jane and Aatos Erkko Foundation (https://jaes.fi). C.W.S is supported by a Vetenskaprådet starting grant from the Swedish Research Council (https://www.vr.se, grant no. 2020-05071). T.A.R. is supported by a Royal Society University Research Fellowship (https://royalsociety.org, grant no. UF130382) and additional funding through the European Molecular Biology Organisation Young Investigator Program (www.embo.org). The funders had no role in study design, data collection and analysis, decision to publish, or preparation of the manuscript.

**Competing interests:** The authors have declared that no competing interests exist.

**Abbreviations:** CM, cytosine methyltransferase; CSM, complete supplement mixture; HGT, horizontal gene transfer; HMM, Hidden Markov model; mtDNA, mitochondrial DNA; MTS, mitochondrial targeting sequence; OGA, Ocean Gene Atlas; ORF, open reading frame; RM, restriction modification; SAG, single-cell amplified genome.

of mosaic distribution across many eukaryotes, consistent with multiple recent gene transfer and loss events [18,19]. Chloroplasts generally exhibit the same reductive evolutionary trends [20], and although horizontal gene transfer (HGT) of a bacterial operon into the chloroplast genome of eustigmatophyte algae (Ochrophyta), including *Monodopsis* and *Vischeria*, has been reported, the functional relevance of this acquisition is not clear [21]. While novel functional genes have entered chloroplast genomes, and genes are sometimes replaced in mitochondrial DNA (mtDNA), the scarcity of gain-of-function transfers identified within mitochondrial genomes makes them intriguing case studies for assessing the functional consequences of such gene acquisitions.

Selfish genetic elements typically serve no function except to replicate themselves [22], even at the cost of host fitness. However, some horizontally transferred selfish elements have been co-opted to perform critical functions in host cells [23]. For example, type II restriction modification (RM) systems can provide host cells with protection from invasion by viruses, plasmids, or other sources of foreign DNA [24]. RM selfish elements work by the coordinated regulation of two enzymes that behave as a type IV toxin–antitoxin system [25]; the restriction endonuclease acts as a "toxin" by cutting DNA at specific recognition sequence motifs, while a methyltransferase acts as an "anti-toxin" by modifying nucleotides at the same recognition sequence, thereby protecting the DNA from cleavage by the cognate endonuclease. If found within an organellar genome, a functional RM system could simply act to protect the genome from invasion by viruses/phages, plasmids, or other sources of foreign DNA. However, RM systems in organelles could also act as a strong "gene drive," ensuring that a single mitochondrial haplotype would quickly sweep to fixation in a sexual population via mitochondrial fusion events.

We have recently explored the content of diverse protist mitochondrial genomes using targeted culture-independent single-cell approaches [26]. This process allowed us to recover the first complete mitochondrial genomes from katablepharids, a group of flagellated heterotrophic unicellular protists, approximately 10 μm in length [27], which are found in both marine and freshwater environments [28,29]. Here, we describe the identification and characterisation of four open reading frames (ORFs) comprising two type II RM selfish elements within katablepharid mtDNA that likely derive from an HGT event into the mitochondrial genome. We report the phylogenetic ancestry of these genes and assess the activity of the encoded enzymes by heterologous expression in *Escherichia coli* and *Saccharomyces cerevisiae*. We suggest that these mitochondrion-encoded proteins may constitute a hitherto undescribed system that could control differential organelle inheritance.

## Results

### Identification of unique restriction modification selfish elements in katablepharid mitochondrial genomes

Our recent initiative to assess mitochondrial genome content using environmentally sampled, protistan, single-cell amplified genome (SAG) sequences resulted in the complete mitochondrial sequence of multiple marine heterotrophic katablepharid protists [26]. The contemporary publication of the complete *Leucocryptos marina* mitochondrial genome confirmed the identity of the SAG-derived mtDNAs as katablepharid [30]. The genomes from the SAG-generated mtDNA and *Leucocryptos* mtDNA were identical in their repertoires of canonical mitochondrial genes, including those encoding tRNAs (**Fig 1A**). They shared synteny throughout the majority of the genome, including near identical intron locations in *rnl*, *cob*, and *cox1* (grey in **Fig 1A**), and retained three unassigned ORFs at identical genomic locations (orange in **Fig 1A**). The regions lacking synteny between the complete genomes encode *atp9*, *rns*, the

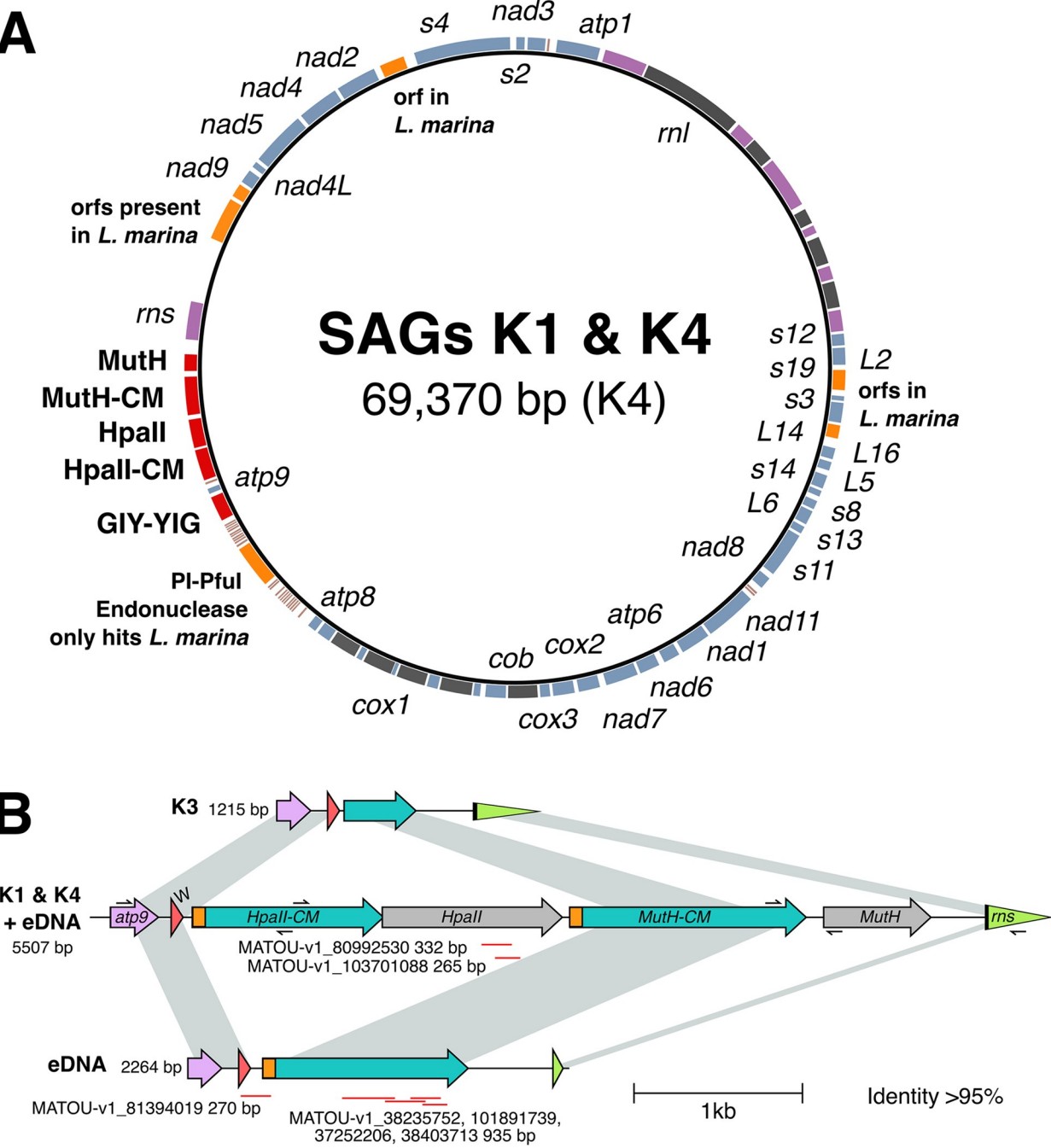

**Fig 1. The katablepharid K1 and K4 mitochondrial genomes encode two tandemly encoded RM systems. (A)** The K4 mitochondrial SAG encodes six ORFs (orange) with homologues in *L. marina* but with no similarity to any other eukaryote. Five ORFs are found in K4 but not in *L. marina* (red). These five genes include a putative GIY-YIG homing endonuclease and two putative RM systems, each consisting of an endonuclease and a CM. Blue, protein-coding genes; purple, rRNA genes; grey, introns. K1 and K4 mitochondrial genome contig data available at https://doi.org/10.6084/m9.figshare.7314728. **(B)** Variation in selfish elements detected in single amplified genomes and eDNA. PCR followed by Sanger sequencing was performed to confirm integration of selfish element genes in K1, K3, and K4 SAGs (primer positions indicated by arrows). PCR of eDNA samples identified one product with high sequence identity (99.7%) to K4 and a shorter (2,264-bp) unique product that is intermediate in length compared to K3 and K4. Red lines represent meta-transcriptome hits (95% to 100% identity) identified from the MATOU transcriptome database (**S1 Data**). Gene name abbreviations: *atp9*, ATP synthase 9 subunit; W, tryptophan tRNA; *HpaII*, HpaII endonuclease; *CM*, cytosine methylase; *MutH*, MutH endonuclease; *rns*, small subunit rRNA gene. Fig 1B was generated using Clinker [31] and modified by hand. ORF colours do not correspond between **Fig 1A** and **1B**. CM, cytosine methyltransferase; eDNA, environmental DNA; ORF, open reading frame; RM, restriction modification; SAG, single-cell amplified genome.

vast majority of *tRNA*s, as well as a variety of unassigned ORFs. Of the eight unassigned ORFs in this region of the SAG-derived katablepharid mitochondrial genome, three had homologues in *L. marina* (orange in **Fig 1A**), but five did not retrieve *L. marina* proteins as top hits using BLASTx searches (red in **Fig 1A**). One of these was a highly divergent GIY-YIG homing endonuclease, two were identified as restriction enzymes, and two were identified as cytosine methyltransferases (CMs) (**Fig 1B**). The restriction enzymes and CMs comprised two tandemly encoded type II RM selfish genetic elements each consisting of a restriction endonuclease and a cognate CM. Specifically, the two katablepharid RMs were identified to be composed of a HpaII restriction endonuclease (Kat-HpaII) and its cognate CM (Kat-HpaII-CM), as well as a MutH/Restriction endonuclease type II (Kat-MutH) with its cognate cytosine-C5 methyltransferase (Kat-MutH-CM) (**Fig 1B**). The Kat-HpaII RM system is flanked by near-identical (152/155-bp) sequences that may reflect the recent integration of this selfish element (shown in orange in **Fig 1B**) into the katablepharid mtDNA.

BLASTP analysis of the putative methyltransferases against the REBASE database (http://rebase.neb.com; accessed January 2021) indicated that Kat-HpaII-CM is likely specific for CCGG DNA recognition sequences, with an *Algibacter* (Flavobacteria) methyltransferase (accession: ALJ03853.1) as the top hit (59% amino acid identity). Furthermore, BLASTP analysis suggested that the Kat-MutH-CM is likely to specifically recognise a GATC DNA recognition sequence, with the top hit a methyltransferase from *Arenitalea lutea* (Flavobacteria) (genome accession: ALIH01000012.1, 74% amino acid identity). Analysis of the putative endonucleases also suggested that Kat-HpaII was specific for a CCGG DNA recognition sequence, with an *Aggregatibacter* (γ-proteobacteria) endonuclease (accession: RDE88890.1) as the top BLASTP hit (36% amino acid identity) and that the Kat-MutH endonuclease may be specific for GATC, recovering as the top BLASTP hit a DNA mismatch repair protein from *Mangrovimonas* (Flavobacteria) (accession: KFB02001.1, 27% amino acid identity). The 2,240-bp region encoding the Kat-HpaII/Kat-HpaII-CM RM system lacks any CCGG motifs, which are expected to occur by chance once every 752-bp in the katablepharid mitochondrion, based on a GC content of 38% for this organeller genome. Taken together, our analysis predicts that the gene products within each katablepharid RM pair target the same recognition sequences.

To confirm that all identified regions were not the result of contamination or genome assembly artefacts, we re-amplified the corresponding region of the mitochondrion from the SAG DNA samples, confirming the four-gene architecture of the selfish element identified was present in 2 samples ("katablepharid 1 (i.e., 'K1')" and "katablepharid 4" (i.e., "K4")) (**Fig 1B**) and that the genes are adjacent to the katablepharid mitochondrial *atp9* and *rns* genes. This PCR analysis also confirmed the existence of a reduced variant (in "katablepharid 3"), which consists of only the amino-terminal region of the MutH-CM gene (**Fig 1B**).

Next, we conducted three separate, targeted PCRs using environmental DNA recovered from parallel marine water samples collected on the same date, and from the same site, as those which contained the individual cells sorted for genome sequencing [26]. These analyses further confirmed that the selfish elements were found adjacent to mitochondrial genes and that the products were not an artefact of multiple displacement amplification as part of the single-cell sequencing pipeline. We also identified an additional contig possessing an intermediate, reduced form of the selfish element gene architecture, which contained only the MutH-CM gene (**Fig 1B**). In total, amplification of the bacteria-like RM systems from mitochondrial genomes was independently sampled five times. Collectively, these results indicate that RM selfish genetic elements have been incorporated into mtDNAs, and that these elements have been subjected to rapid evolutionary change, including gene loss/ORF reduction.

To explore if the identified selfish element genes are expressed, we interrogated a collection of marine meta-transcriptome data publicly available at the Ocean Gene Atlas (OGA; available

at http://tara-oceans.mio.osupytheas.fr/ocean-gene-atlas/). We identified a number of eukaryotic transcripts from geographically diverse marine sampling sites with high percentage nucleotide identity to the *kat-HpaII*, *kat-HpaII-CM*, and *kat-MutH-CM* genes (**S1 Data**), thereby demonstrating that this selfish genetic element is expressed from the katablepharid mitochondrial genome. Regions identified in the MATOU_v1_metaT transcriptome database with >95% identity are shown in **Fig 1B**. Interestingly, two of the OGA RNAseq derived contigs that showed >99% nucleotide identity to the Kat-HpaII mitochondrial gene were composed of sequence reads sampled from multiple sites in the Pacific, Southern Atlantic, and Indian Oceans, and the Mediterranean Sea. These samples included both "surface" and "deep chlorophyll maximum" samples. These results suggest that the Kat-HpaII endonuclease is transcriptionally active across a wide range of ocean environments. Furthermore, the OGA transcript sequences included one contig that traverses the *trnW* gene and the repetitive flanking sequence upstream of Kat-HpaII-CM/Kat-MutH-CM gene cluster, indicative of mitochondrial co-transcription.

## Katablepharid mitochondrial RM system has flavobacterial ancestry

To explore the phylogenetic ancestry of the selfish genetic element, we conducted phylogenetic analysis using Bayesian and maximum-likelihood approaches, with a focus upon the complete Kat-HpaII-CM and Kat-HpaII found in the K4 assembly [26], as Kat-MutH-CM and Kat-MutH were found to have no detectable function (discussed below). The phylogeny of the restriction endonuclease showed the mitochondrial genes branching basally to the flavobacteria, with weak bootstrap support (**S1A Fig**). In contrast, phylogenetic analysis of Kat-HpaII-CM (**S1B Fig**) and the concatenated alignment of both Kat-HpaII and Kat-HpaII-CM demonstrated strong bootstrap support (**Fig 2A**) for the mitochondrial selfish genetic element branching within a clade of flavobacteria. There is currently no evidence that flavobacteria, or genetic material derived from the flavobacteria, played a role in the origin of the eukaryotes or the mitochondria [32,33]. Furthermore, the katablepharid SAG assemblies contained no obvious contaminating flavobacteria-like sequences (**S2 Fig**). As such, we conclude that the selfish genetic element has been transferred to the mitochondrial genome from a donor species that branches within, or close to, the flavobacteria. Our phylogenetic analysis demonstrated that the selfish genetic element represented a longer branch in the phylogeny (**Fig 2A**), suggesting evolutionary scenarios consistent with invasion of the mtDNA genome, such as positive and/or relaxed selection.

To further explore the nature of sequence evolution associated with this HGT event, we calculated the codon usage frequencies of the Kat4-HpaII RM, the conserved protein-coding gene repertoire of the Kat4 mtDNA, and the *Algibacter*-HpaII RM. Using Fisher exact tests, we demonstrated that codon usage was significantly different for 14 amino acids when comparing the Kat4 complement of unambiguously ancestral Kat4 mitochondrial proteins and the *Algibacter*-HpaII RM. In contrast, the Kat4-HpaII RM represents an intermediate, exhibiting six amino acids with codon usage differing from the Kat4 mtDNA (again sampling all unambiguously ancestral Kat4 mitochondrial proteins), and seven amino acids with codon usage differing from that of *Algibacter*-HpaII RM (**Fig 2B**; see **S2 Data** for raw data). These data are consistent with the hypothesis that the Kat4-HpaII RM is in the process of amelioration towards the sequence characteristics of the host mtDNA. Such changes may also be, in part, a driver and/or consequence of the accelerated evolutionary rate indicated by the long branch the katablepharid selfish element forms in the phylogenetic trees (**Fig 2A**).

## Confirmation of a functional mitochondrial katablepharid methyltransferase

To explore the function of the selfish element and to test if it has undergone pseudogenisation, we cloned the Kat-HpaII-CM and its closest bacterial homologue in terms of sequence

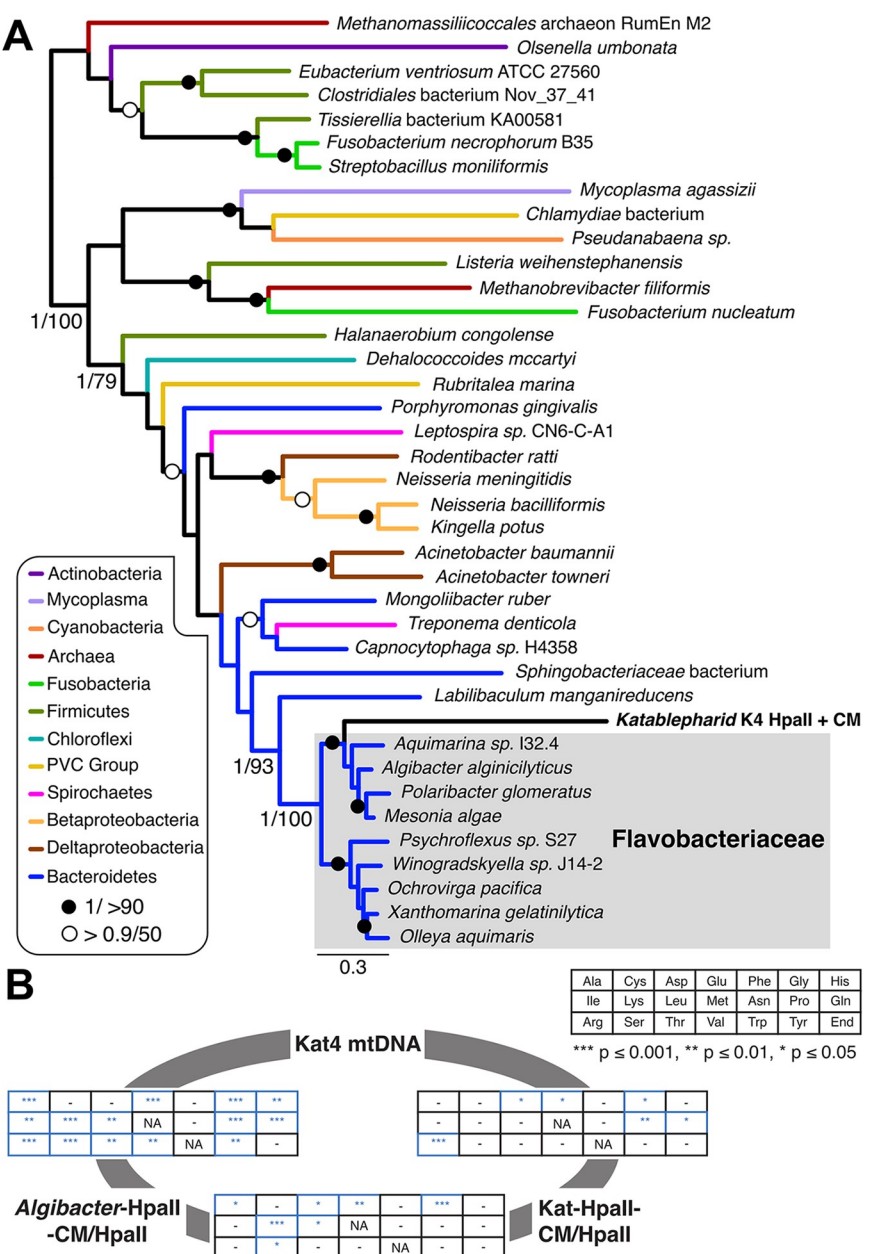

**Fig 2. Phylogenetic reconstruction of concatenated genes encoded by katablepharid RM selfish elements and comparison of codon frequencies between katablepharid and *Algibacter* complements. (A)** A concatenated phylogeny was reconstructed using sequences from K4 and 38 prokaryotic species containing tandemly encoded HpaII-CM and HpaII proteins. The concatenation resulted in an alignment length of 767 amino acid positions. Support values are posterior probabilities calculated using MrBayes v3.2.6 [34] and 100 bootstrap replicates using RAxML v8.2.10 [35] and reported as MrBayes/RAxML. The MrBayes topology is shown. Species phyla are indicated as differently coloured branches as depicted inset. Bipartitions with support lower than 0.9/50 are unlabelled. For individual trees of HpaII and HpaII-CM, see **S1 Fig**. Alignment data available at http://doi.org/10.6084/m9.figshare.c.5336963. **(B)** Pairwise comparisons of sets of alternative codon frequencies for Kat4-HpaII-CM/HpaII, *Algibacter*-HpaII-CM/HpaII, and the conserved protein-coding gene repertoire of the Kat4 mtDNA. Pairwise comparisons are shown in the respective grids. The key shows a grid with the corresponding amino acids. Results for Fisher exact tests comparing codon usage for each amino acid are shown in the tables. Asterisks denote significantly different codon usage, "-" indicates no significant difference in codon frequencies, and "NA" indicates methionine and tryptophan, which were not tested as these amino acids are encoded by a single codon. Grids are placed on a grey circle between the three compared gene sets to identify the results of each pairwise comparison. Raw data available in **S2 Data**. mtDNA, mitochondrial DNA; RM, restriction modification.

identity, *Algibacter* HpaII-CM (Alg-HpaII-CM), into plasmid pACYC184 and expressed them in *E. coli* Top10 cells. This *E. coli* strain does not contain any methyltransferases that target the putative HpaII-CM recognition sequence (CCGG), but instead expresses Dcm methylase, which methylates the second cytosine residue in CCWGG [36], and Dam methylase, which methylates adenine residues in the sequence GATC [37]. The HpaII-CM-expressing *E. coli* Top10 strains were cultured for 16 hours alongside a control strain harbouring an empty plasmid. These plasmids were then extracted, linearised and subject to bisulfite conversion, a process that converts cytosine nucleotides to uracil but does not alter methylated 5-methylcytosines (5-mC). A 299-bp region of each plasmid was PCR amplified and sequenced. In the plasmid from the control strain, all CCGG sites appeared as TTGG in sequencing chromatograms, whereas plasmids sequenced from strains expressing Kat-HpaII-CM or Alg-HpaII-CM contained TCGG sites (**Fig 3A**), demonstrating the ability of Kat-HpaII-CM to methylate CCGG sequences at the second cytosine base.

To confirm Kat-HpaII-CM function, we transformed the plasmid expressing this katablepharid sequence into *E. coli* strain DH5α. This strain encodes McrA, which cleaves DNA sequences methylated at the second cytosine of the HpaII recognition motif ($C^{me}CGG$) [38]. As a control, we also transformed Kat-HpaII-CM into the *mcrA- E. coli* Top10 strain. Comparisons of transformation efficiency confirmed that the katablepharid methyltransferase is highly toxic in the *mcrA+ E. coli* DH5α background (**Fig 3B**), further demonstrating that Kat-HpaII-CM encodes a functional enzyme which methylates CCGG sites.

Next, we performed bisulfite conversion and sequencing experiments using Kat-MutH-CM. Here we targetd an alternative 233-bp region of the plasmid to enable detection of potential methylation at GATC sites. However, we found no evidence of any methylation at GATC, or other sites, by Kat-MutH-CM, suggesting that this enzyme may have lost its GATC specific catalytic activity, requires additional factors for function, or has evolved an alternative function.

## Confirmation of a functional mitochondrial katablepharid endonuclease

To explore the function of the candidate endonucleases, we cloned the putative *Algibacter* HpaII endonuclease (Alg-HpaII), the katablepharid MutH-like endonuclease (Kat-MutH), and Kat-HpaII into a pBAD expression vector, transformed these plasmids into *E. coli*, and compared culture growth for each resulting strain. The strain-expressing Kat-MutH showed no evidence of toxicity, demonstrating a similar growth dynamic to the *E. coli* strain containing only the empty vector (**S3 Fig**), and the functions of Kat-MutH-CM and Kat-MutH were not pursued further. In contrast, cultures of strains expressing the Kat-HpaII and Alg-HpaII grew slowly, consistent with these genes encoding functional endonucleases that constitute a bona fide "toxin" (**Fig 3C**). The katablepharid HpaII showed a greater potency during these experiments when compared to the *Algibacter* HpaII. In order to explore if the Kat-HpaII and Kat-HpaII-CM function as a toxin/anti-toxin pair, we co-expressed these two proteins. This demonstrated that Kat-HpaII-CM was able to partially reverse the effects of Kat-HpaII expression in *E. coli* (**Fig 3D**). Subsequent experiments that increased the expression of the Kat-HpaII enzyme by removal of an additional ATG at the 5′ of the sequence led to this rescue being perturbed (**S4 Fig**), suggesting that differences in the relative expression of the toxin/antitoxin can determine the degree of toxicity.

## Targeting of HpaII-CM and HpaII to yeast mitochondria confirms methyltransferase and endonuclease activities

To further explore the potential roles of Kat-HpaII-CM and Kat-HpaII in katablepharid mitochondria, we targeted each protein to the mitochondria of *S. cerevisiae* cells using an amino-

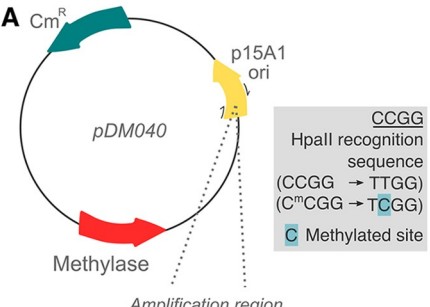

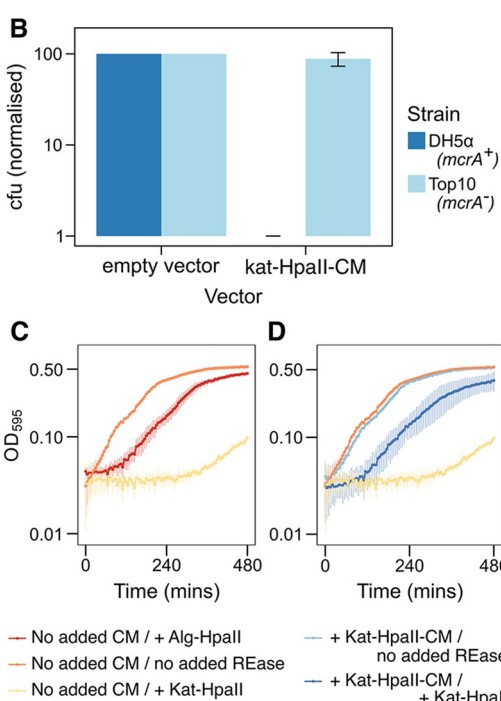

**Fig 3. Heterologously expressed katablepharid HpaII-CM and HpaII are catalytically active. (A)** Bisulfite conversion to identify 5-methylcytosine modification by the katablepharid HpaII-CM. Schematic of bisulfite conversion protocol to assess 5-methylcytosine modifications. Plasmids were purified from *E. coli* Top10 and subjected to bisulfite conversion to convert cytosine to uracil (replaced with thymine during PCR), while 5-methylcytosines (5-mC) remain unaffected. Moreover, 5-mC residues were detected within the amplification region when the katablepharid HpaII-CM was present on plasmid pDM040. Notably, each methylated site (indicated in blue) was located at CCGG, an HpaII recognition sequence (underlined). **(B)** Transformation efficiency of *E. coli* strains when transformed with putative katablepharid HpaII-CM. Transformation efficiency of *E. coli* DH5α and Top10 strains when transformed with empty vector control (pACYC184) or vector containing the katablepharid putative methyltransferase coding sequence (pDM40). Experiments were performed from a minimum of three independent competent cell batches, and CFUs were enumerated and normalised to the positive control (pACYC184) within each batch. These data demonstrate that the katablepharid HpaII methyltransferase is toxic in *E. coli* DH5α (*mcrA+*), but not in Top10 (*mcrA-*). Error bars represent one standard deviation from the mean. Underlying data in S3 **Data**. **(C)** Growth of *E. coli* Top10 cells with combinations of plasmids containing putative katablepharid methyltransferase (CM +), katablepharid HpaII (*Kat* HpaII), and Algibacter HpaII (*Alg* HpaII) genes or the corresponding empty vectors ("no added CM" or "no added REase" [restriction endonuclease]). Duplicate cultures from independent transformants were grown for eight hours under Amp/Cm selection, induced with 0.0004% arabinose, at 37˚C, and growth was assessed by measuring OD₅₉₅ at five-minute intervals. The strain lacking the endonuclease showed typical *E. coli* growth, while addition of either the Algibacter (*Alg*) endonuclease or Katablepharid (*Kat*) endonuclease to the strain lacking the

methyltransferase caused toxicity. **(D)** Addition of the katablepharid methyltransferase (CM+) rescued this toxicity to near control levels of growth (controls transposed from **C**). Error bars represent one standard deviation from the mean. Underlying data for C and D in **S4 Data**. CFU, colony-forming unit; CM, cytosine methyltransferase.

terminal Su9 mitochondrial targeting sequence (MTS) from *Neurospora crassa* [39]. Constructs also contained a carboxyl-terminal GFP tag to allow confirmation of mitochondrial localisation, and proteins were controlled by a galactose-inducible promoter to allow temporal induction of gene expression. Following induction of su9(MTS)-Kat-HpaII-CM-GFP, we sequenced a region of the mitochondrial *COX1* gene following bisulfite conversion. As seen following heterologous expression in *E. coli*, CCGG sites of mtDNA were methylated, indicating that Kat-HpaII-CM could function with in the context of a mitochondrial matrix (**Fig 4A**). We also sequenced this region of the *COX1* gene from a *S. cerevisiae* strain lacking the su9 (MTS)-Kat-HpaII-CM-GFP plasmid and demonstrated that these residues are not methylated in wild-type cultures.

We then assessed whether su9(MTS)-Kat-HpaII-CM-GFP would be recruited to mtDNA by staining mitochondrial nucleoids with DAPI [40] and we found that the katablepharid methyltransferase co-localised with punctate DAPI foci (**Fig 4B**). When su9(MTS)-Kat-HpaII-GFP was targeted to mitochondria, this protein was also found in puncta, yet a co-localised the DAPI signal appeared absent, indicating that the mtDNA has likely been degraded (**Fig 4B**). This ability of Kat-HpaII to damage mtDNA was supported by an increase in the

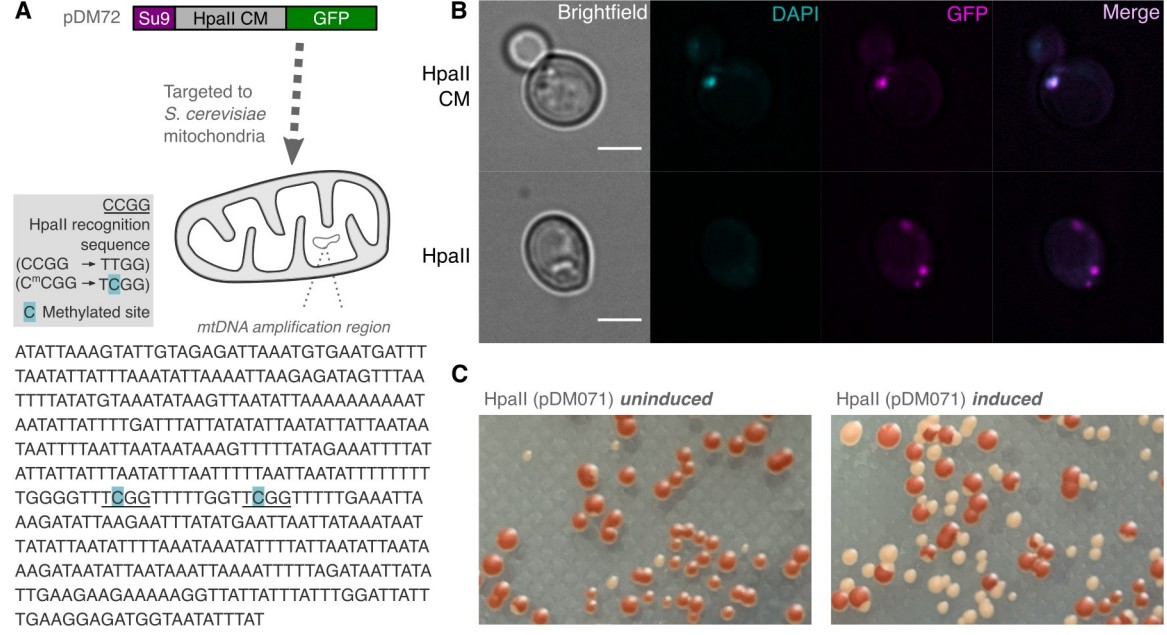

**Fig 4. Katablepharid HpaII endonuclease and methyltransferase function in *S. cerevisiae* mitochondria.** **(A)** Bisulfite conversion and subsequent sequencing to confirm targeting of a functional HpaII-CM to yeast mitochondria. Schematic of bisulfite conversion protocol to assess 5-methylcytosine modifications after induction of the katablepharid HpaII-CM from plasmid pDM072. 5-mC residues were detected within the amplification region of the *cox1* gene. Each methylated site (indicated in blue) was located at the HpaII recognition sequence (underlined). **(B)** Evaluation of fluorescence for GFP-tagged HpaII-CM and HpaII, in conjunction with DAPI-labelled mtDNA. The HpaII-CM showed co-localisation with DAPI, while HpaII showed an absence of a DAPI focus, indicative of a lack of mtDNA, which is likely to have been a result of endonuclease function and DNA degradation. Scale bar = 3 μm. **(C)** HpaII expression causes petite formation. Formation of petite colonies (white; due to a lack of an electron transport chain function) after the induction of HpaII (right), in comparison to an uninduced strain (left). CM, cytosine methyltransferase.

formation of petite colonies after su9(MTS)-Kat-HpaII-GFP induction (**Fig 4C**). Taken together, our results indicate that the Kat-HpaII-CM and Kat-HpaII are able to function within mitochondria to methylate and degrade mtDNA.

## Discussion

Here, we have revealed the integration of a functional type II RM system into the mitochondrial genome of a katablepharid protist. We confirm that Kat-HpaII and Kat-HpaII-CM are active when expressed in both prokaryotic and eukaryotic cells, and we provide evidence demonstrating a toxin/antitoxin functional relationship between these two proteins.

Why would this active RM selfish element reside within a mitochondrial genome? We suggest several possible evolutionary scenarios. First, the katablepharid type II RM selfish element could simply represent a recent invasion that is of no functional or evolutionary consequence for its katablepharid host. We do see evidence of RM system degeneration within some of the katablepharid mitochondrial genome sequences sampled, including loss of the HpaII/HpaII-CM genes, as well as the presence of a partial MutH-CM gene, suggesting that selection for maintenance of this selfish element is patchy, and loss is tolerated.

Second, the Kat-HpaII and Kat-HpaII-CM system may protect mitochondria from foreign DNA. In bacteria, RM selfish elements are thought to function as a defence against foreign unmethylated DNA [41], such as viruses/phages and plasmids, which are also known to invade mitochondria [42,43]. However, any fitness benefit to cells harbouring these elements related to this function would be conditional upon regular exposure to foreign sources of DNA. Consistent with this proposition, we detected evidence of expression of the Kat-HpaII gene from multiple oceanic environments, implicating a wide biogeographic distribution of active gene transcription.

Third, and most intriguing among these possibilities, this RM element could act to bias the spread of the host mtDNA within the katablepharid population. Previous studies indicate that selfish mitochondrial mutants spread rapidly in a sexual population, whereas asexual populations are relatively protected from similar patterns of invasions [44–47]. Furthermore, mitochondrial fusion is documented in many eukaryotes [48,49]. Thus, crosses of Kat4-HpaII RM + and RM− individuals would, hypothetically, initially result in a mixed population of mitochondria. After mitochondrial fusion, RM+ mtDNA could lead to digestion of unprotected, RM− mtDNA, leading to selfish element-mediated, biased inheritance of RM+ mtDNA and the rapid spread of this mitochondrial haplotype. To further explore this possibility, we sought genes which putatively encode meiosis components in our four katablepharid SAGs and identified gene fragments of six meiosis-associated proteins (MSH5, XRCC3, DMC1, SPO11, Brambleberry, and SNF2; see **S5 Data**) in K2/K4. Our results suggest that katablepharids, like most eukaryotes, contain meiosis-specific genes (e.g., SPO11; MSH5 [50]), and may be capable of sexual reproduction, although sex has not been directly observed in this lineage [51]. These SAGs are incomplete and extremely fragmented [26], so confirmation of meiotic genes in this clade will require additional data.

Biased or strictly uniparental inheritance of cytoplasmic organelles is a consistent trend across diverse eukaryotic groups and has multiple, independent origins [52]. Therefore, the invasion of organellar genomes by RM selfish elements may constitute a hitherto unrecognised mechanism for gene drive that enables differential inheritance of mitochondrial genomes, independent of, and potentially overriding, direct nuclear control. While methylation/nuclease functions may contribute to the uniparental inheritance of chloroplasts in *Chlamydomonas* [53,54], the mechanisms in this system are unclear, and the genes responsible have not been reported to be a consequence of an HGT invasion event. Furthermore, the invasion of selfish genetic elements, based on toxin–antitoxin function, into organellar genomes has been

predicted [55], although, until now, undiscovered. This prediction sets out that selfish genetic elements will take up important roles in inter-organellar genome conflict [55], and it is possible that the RM system identified here has become a weapon deployed in a war between mitochondrial genomes in the katablepharid lineage.

## Materials and methods

### Phylogenetic analysis of restriction modification selfish elements encoded in katablepharid mitochondrial genomes

To determine the origins of the katablepharid mitochondrion-encoded RM selfish element, we collected putative homologues from the NCBI *nr* database using katablepharid HpaII (Kat-HpaII) and HpaII-CM (Kat-HpaII-CM) as queries. The top hits were predominantly from the Flavobacteriaceae, suggesting that the katablepharid RM originated within this group. To confirm the phylogenetic origins of the katablepharid RM system, we collected protein sequences from diverse bacterial phyla (Autumn 2020) that encoded HpaII and HpaII-CM in tandem, then reconstructed single-gene and concatenated phylogenies. HpaII and HpaII-CM orthologues were aligned with MUSCLE [56] and manually trimmed using Mesquite v.2.75 [57]. The two-gene concatenation was performed by hand in Mesquite v.2.75. Phylogenetic tree reconstructions were performed using MrBayes v.3.2.6 for Bayesian analysis [34] using the following parameters: prset aamodelpr = fixed (WAG); mcmcngen = 2,000,000; samplefreq = 1,000; nchains = 4; startingtree = random; and sumt burnin = 250. Splits frequencies were checked to ensure convergence. Maximum-likelihood bootstrap values (100 pseudoreplicates) were obtained using RAxML v.8.2.10 [35] under the LG model [58].

### Analysis of codon usage

Codon usage frequencies of the proteins encoded by the Kat4 and Algibacter HpaII and HpaII-CM selfish elements, as well as the unambiguously ancestral Kat4 mitochondrial proteins [32], were determined using the Sequence Manipulation Suite server [59]. Amino acid codon usage frequencies were compared using a Fisher exact test in R (version 1.3.1073) [60].

### Identification of katablepharid-related restriction modification systems in metagenomic databases

All four genes of the two selfish elements were BLASTN searched against the OGA [61] (searched December 2020, tool available at http://tara-oceans.mio.osupytheas.fr/ocean-gene-atlas/) OM-RGC_v2_metaT (prokaryote) and MATOU_v1_metaT (eukaryote) transcriptome databases. Only hits of over 100-bp in length with DNA identity scores in excess of 95% were retained for further analysis (see **S1 Data**).

### Identification of putative meiosis protein encoding genes in katablepharid SAGs

Hidden Markov models (HMMs) corresponding to meiosis-associated proteins [62,63] were retrieved from Pfam, PNTHR, EGGNOG, and TIGR databases via InterPro (https://www.ebi.ac.uk/interpro/; November 2020); see **S5 Data** for accession numbers. These HMMs were used as queries against a 6-frame translation of the Katablepharid SAGs (K1: sample 11B_35C, K2: 11H_35C, K3: 5F_35A, K4: 6E_35B; https://figshare.com/articles/dataset/Single_Cell_Genomic_Assemblies/7352966) using hmmsearch with an e-value (-E) cut-off of 0.1, with all other parameters at default. The nucleotide sequences from resulting hits were used as queries against the nonredundant (*nr*) database (November 2020) using BLASTX [64] to allow for

intron read-through. If the majority of the top hits against the *nr* database corresponded to the same meiosis-associated protein, then the sequence was included in **S5 Data**.

## PCR confirmation of katablepharid restriction modification selfish elements

To validate the presence of the RM system on the katablepharid mitochondrial genome assembly, and to further assess the katablepharid mitochondrial RM diversity, we conducted PCR, using a range of templates: (i) a katablepharid SAG DNA from Wideman and colleagues [26]; and (ii) DNA extracted from a water sample taken at a depth of 20 m from the same site, the Monterey Bay Aquarium Research Institute time-series station M2, and on the same date, as the single-cell isolations [26]. PCR amplifications were performed using Phusion polymerase (New England Biolabs, Ipswich, MA) and the primers detailed in **S1 Table**. Each 25 µl reaction contained 200 nM of each primer, 400 nM dNTPs, and 1 ng template DNA. Cycling conditions were 2 minutes at 98°C followed by 30 cycles of 10 seconds at 98°C, 20 seconds at 64.3°C, 2 to 3 minutes at 72°C, and a final extension of 7 minutes at 72°C. PCR products were purified (Gene-Jet PCR Purification Kit, Thermo Fisher Scientific, Waltham, MA), adenosine-tailed using GoTaq Flexi DNA polymerase (Promega, Madison, WI), and cloned into pSC-A-amp/kn using a StrataClone PCR Cloning Kit (Agilent Technologies, Santa Clara, CA). Plasmids were then Sanger sequenced using T7/T3 primers or the original PCR primers (Eurofins Genomics, Germany), with additional internal sequencing reactions performed when necessary.

## Plasmid construction

Sequences were codon optimised for *E. coli* or *S. cerevisiae* expression and synthesised *de novo* (Synbio Tech, NJ). For *E. coli* expression, putative methyltransferases were cloned into the BamHI/SalI sites of the low copy vector pACYC184 (New England Biolabs) with an upstream Shine-Dalgarno consensus sequence (5′-AGGAGG-3′), and putative endonucleases were cloned into the PstI/HindIII sites of pBAD HisA (Thermo Fisher Scientific). For expression of proteins in *S. cerevisiae*, each ORF was fused to an amino-terminal Su9 pre-sequence from *N. crassa* for targeting to the mitochondrion and to a carboxyl-terminal GFP tag for visualisation by fluorescent microscopy. Kat-HpaII-CM and Kat-HpaII were cloned into the BamHI/KpnI sites of pYX223-mtGFP and pYES-mtGFP plasmids, respectively [39]. All plasmid constructs are detailed in **S2 Table**.

## *E. coli* transformation and proliferation assays of strains expressing components of the RM system

Plasmids containing putative methyltransferase and endonuclease genes were transformed into chemically competent *E. coli* Top10 (*dcm+ dam+*, *mcrA-*) or DH5α (*dcm+ dam+*, *mcrA+*). Where transformations into DH5α were unsuccessful, independent triplicate transformations were performed into both Top10 and DH5α to assess strain-specific incompatibility. This was achieved by performing transformations where equal concentrations (50 ng) of pDM040 or pACYC184 (empty vector control) were added to each competent cell aliquot, before plating onto LB $Cm_{45}$, incubating at 37°C for 16 hours, then counting colony forming units.

To assess proliferation of each *E. coli* Top10 strain, duplicate cultures were grown for 16 hours at 37°C (200 rpm shaking) in LB $Amp_{50}$ $Cm_{45}$ before being diluted to $OD_{600}$ 0.1 in the same medium. A total of 100 µL of each culture was inoculated into a 96-well plate and incubated at 37°C with 200 rpm double-orbital shaking in a BMG FLUOstar Omega Lite instrument. Proliferation was assessed by measuring $OD_{595}$ at 5-minute intervals for 480 minutes. All *E. coli* replicates were from independent transformants.

## Bisulfite conversion to assess for methylase activity

To assay for 5-methylcytosine (5-mC) methyltransferase activity, *E. coli* Top10 strains with a pACYC184 vector containing Kat-HpaII-CM (pDM040), Kat-MutH-CM (pDM042), or *Algibacter* methyltransferase (Alg-HpaII-CM) (pDM041) were grown for 16 hours at 37˚C (200 rpm shaking) in LB Cm$_{45}$. Plasmids were extracted using a GeneJet Plasmid Miniprep kit (Thermo Fisher Scientific), linearised using HindIII to avoid supercoiling, then gel extracted (Promega Wizard SV Gel and PCR Clean-Up System). Linear plasmids were subjected to bisulfite conversion using the EpiMark Bisulfite Conversion Kit (New England Biolabs), following the manufacturer's instructions. A 299-bp region of each plasmid was amplified with primers pACYC184_5mC_F and pACYC184_5mC_R2 to assess CCGG methylation, and a 233-bp region was amplified with primers pACYC184_region2_5mC_F2/R2 to assess GATC methylation. Both primer pairs (S1 Table) were designed to amplify bisulfite-converted DNA. A total of 25 μl reactions containing 1x GoTaq G2 Hot Start Green Master Mix (Promega), 1 μM each primer, and 1 μL of 100-fold diluted plasmid template were used, with the following cycling conditions: 2 minutes at 94˚C, followed by 35 cycles of 15 seconds at 94˚ C, 30 seconds at 50˚C, and 30 seconds at 72˚C, then a final extension of 5 minutes at 72˚C. PCR products were then purified (Promega Wizard SV Gel and PCR Clean-Up System) and sequenced on both strands (Eurofins Genomics) to identify bases which remained as cytosines, indicative of a 5-mC modification at this site.

To assess mtDNA methylation in *S. cerevisiae* cells, 1 mL of culture was purified using a Promega Wizard genomic DNA purification kit, following the manufacturer's instructions for isolating genomic DNA from yeast. Bisulfite conversion was performed as above, with 500 ng of genomic DNA used in each reaction. Primers cox1_bisulfite_F and cox1_bisulfite_R (S1 Table) were then used to amplify a 443-bp region of the mitochondrial *cox1* gene using GoTaq Hot Start Master Mix (Promega). Each 50 μL reaction contained 500 nM each primer and cycling conditions were as described above. PCR products were purified using a Wizard PCR clean-up kit (Promega) before sequencing (Eurofins Genomics).

## GFP localisation of heterologously expressed RM system components in yeast using spinning disc confocal microscopy

Plasmids pDM071, encoding su9(MTS)-Kat-HpaII-GFP, or pDM072, encoding su9(MTS)-Kat-HpaII-CM-GFP (S2 Table), were transformed into competent *S. cerevisiae* BY4742 cells, using the method described by Thompson and colleagues [65], and selected on Scm-ura agar [0.69% yeast nitrogen base without amino acids (Formedium), 770 mg L$^{-1}$ complete supplement mix (CSM) lacking uracil (Formedium), 2% (wt/vol) glucose, and 1.8% (wt/vol) Agar No. 2 Bacteriological (Neogen, Lansing, MI)], or Scm-his agar (containing CSM-histidine in place of CSM-uracil), respectively. Cells were grown for 16 hours in Scm-his or Scm-ura plus 2% glucose at 30˚C, diluted 10-fold, and induced in fresh media containing 2% galactose (instead of glucose) for 12 hours. Cells were then harvested by centrifugation and resuspended in sterile water containing 1 μg mL$^{-1}$ DAPI for 30 minutes, which preferentially labels mtDNA in the absence of fixation [40]. Cells were then observed by spinning-disc confocal microscopy using an Olympus IX81 inverted microscope affixed with a CSU-X1 Spinning Disk unit (Yokogawa, Tokyo, Japan) and 405 nm/488 nm lasers.

## Assessment of petite formation in *S. cerevisiae*

A *S. cerevisiae* W303 derivative, CAY169 [66] harbouring plasmid pDM071 (HpaII), was grown to mid-logarithmic phase in Scm-ura [0.69% yeast nitrogen base without amino acids (Formedium), 770 mg L$^{-1}$ CSM lacking uracil (Formedium), 2% (wt/vol) glucose, 20 mg L$^{-1}$

adenine sulfate]. Cells were pelleted and resuspended in fresh media containing 2% galactose (instead of glucose) for 12 hours to induce expression of su9(MTS)-Kat-HpaII-GFP, then plated on Scm-ura agar (as above, containing glucose as the sole carbon source). Plates were incubated for 3 days at 30°C and imaged to assess the formation of petite colonies following the pulse of mitochondrial Kat-HpaII expression.

## Supporting information

**S1 Fig. Phylogenetic reconstruction of HpaII (A) and HpaII-CM (B) encoded by the katablepharid mitochondrial genome.** Phylogenies were reconstructed using sequences from K4 and 38 prokaryotic species containing tandemly encoded HpaII and HpaII-CM proteins, resulting in alignments of 352 and 415 positions, respectively. Support values are posterior probabilities calculated using MrBayes v3.2.6 [34] and 100 bootstrap replicates using RAxML v8.2.10 [35] and reported as MrBayes/RAxML. The MrBayes topology is shown. Bipartitions with support lower than 0.9/50 are unlabelled. Alignment data available at http://doi.org/10.6084/m9.figshare.c.5336963. CM, cytosine methyltransferase.
(TIFF)

**S2 Fig. Katablepharid single amplified genomes contain no strong signal for bacterial contaminants.** The contigs assigned to bacteria were low; as such, we have shown assignment only at the taxonomic level of "Bacteria" and have not shown lower taxonomic divisions. Blob-plots were generated using BLOBTOOLS [67] for the 3 SAGs (K4, K1, and K3) that mapped to katablepharids using contigs >1,000-bp. None of the contigs with best BLAST hits to bacteria were related to flavobacterial sequences. SAG contig data can be found at https://doi.org/10.6084/m9.figshare.7352966. SAG, single-cell amplified genome.
(TIFF)

**S3 Fig. Growth assay of *E. coli* Top10 expressing putative katablepharid MutH-like endonuclease.** Growth of *E. coli* Top10 cells with pBAD plasmid containing putative MutH-like endonuclease (MutH) genes, or the corresponding empty vector. Cultures from three independent transformants were grown at 37°C for eight hours under Amp/Cm selection, induced with 0.1% arabinose, and growth was assessed by measuring $OD_{595}$ at 5-minute intervals. This demonstrates that addition of the MutH-like endonuclease does not cause *E. coli* toxicity. Error bars represent one standard deviation from the mean. Underlying data in **S6 Data**.
(TIFF)

**S4 Fig. Perturbation of HpaII rescue by the katablepharid HpaII-CM after modifying the endonuclease constructs.** Removal of the start codon of the ORF from the endonuclease pBAD expression vectors (leaving only the start codon encoded by the vector) resulted in the katablepharid HpaII-CM no longer offering protection against the katablepharid HpaII endonuclease (Kat HpaII) (**A**). However, the katablepharid HpaII-CM was still able to protect against the *Algibacter* HpaII endonuclease (Alg HpaII) (**B**), these results point towards a necessary concentration/function minimum requirement for rescue of katablepharid HpaII endonuclease. Error bars represent one standard deviation from the mean of three independent *E. coli* transformants. Underlying data in (**A**) **S7 Data** and (**B**) **S8 Data**. CM, cytosine methyltransferase; ORF, open reading frame.
(TIFF)

**S1 Table. Primers used in this study.**
(PDF)

**S2 Table. Plasmids used in this study.** Note that all functions described are putative and constructs are codon-optimised for expression in *E. coli*, unless otherwise stated.
(PDF)

**S3 Table. Protein sequences.** Amino acid sequences for all proteins used in this study.
(PDF)

**S1 Data. Data from BLASTN search against the OGA.** Table of all BLASTN hits over 100-bp, with identity scores over 95%. Hits highlighted in grey align with the near-identical regions that flank the HpaII/HpaII-CM RM system; as such, these align with the 5′ of both *hpaII CM* and *mutH CM*, so cannot be attributed to either gene. CM, cytosine methyltransferase; OGA, Ocean Gene Atlas.
(XLSX)

**S2 Data. Codon usage frequency data for proteins encoded by the Kat4 and *Algibacter* HpaII and HpaII-CM selfish elements, and the ancestral Kat4 mitochondrial proteins.** Frequency of each amino acid codon for each of the Kat4-HpaII RM, the conserved protein-coding gene repertoire of the Kat4 mtDNA, and the *Algibacter*-HpaII RM. A Fisher exact test was used to compare codon usage frequencies for each pairwise comparison; *p*-values are displayed beneath each amino acid. CM, cytosine methyltransferase; mtDNA, mitochondrial DNA.
(XLSX)

**S3 Data. Underlying data for Fig 3B.** All data used to create Fig 3B; "sd" = standard deviation from the mean.
(XLSX)

**S4 Data. Underlying data for Fig 3C and 3D.** All data used to create Fig 3C and 3D; "sd" = standard deviation from the mean.
(XLSX)

**S5 Data. Sequence data of putative meiosis-associated genes identified in katablepharid SAGs.** Nucleotide and amino acid sequences for putative meiosis-associated proteins. Interruptions in ORFs strongly suggest the presences of introns. Each protein was identified using the indicated HMM and manually investigated. For each entry, the putative nucleotide sequence that could be confidently identified with BLAST is shown. The accession number for the parent contig of each sequence is provided; these can be accessed at https://figshare.com/articles/dataset/Single_Cell_Genomic_Assemblies/7352966. HMM, Hidden Markov model; ORF, open reading frame; SAG, single-cell amplified genome.
(XLSX)

**S6 Data. Underlying data for S3 Fig.** All data used to create S3 Fig; "sd" = standard deviation from the mean.
(XLSX)

**S7 Data. Underlying data for S4A Fig.** All data used to create S4A Fig; "sd" = standard deviation from the mean.
(XLSX)

**S8 Data. Underlying data for S4B Fig.** All data used to create S4B Fig; "sd" = standard deviation from the mean.
(XLSX)

## Acknowledgments

The authors would like to thank Dayana Salas-Leiva for helpful advice about meiosis and Alexandra Worden's Group for assistance in provision of samples for the initial study [26]. pYX223-mtGFP and pYES-mtGFP were a gift from Benedikt Westermann (Addgene plasmid # 45051/45053). This project was devised at a joint laboratory meeting for which the Wissenschaftskolleg zu Berlin provided accommodation and EMBO provided funding for travel.

## Author Contributions

**Conceptualization:** David S. Milner, Jeremy G. Wideman, Courtney W. Stairs, Cory D. Dunn, Thomas A. Richards.

**Data curation:** David S. Milner, Jeremy G. Wideman, Courtney W. Stairs, Thomas A. Richards.

**Formal analysis:** David S. Milner, Jeremy G. Wideman, Thomas A. Richards.

**Funding acquisition:** Thomas A. Richards.

**Investigation:** Jeremy G. Wideman, Courtney W. Stairs, Thomas A. Richards.

**Methodology:** David S. Milner, Jeremy G. Wideman, Cory D. Dunn, Thomas A. Richards.

**Project administration:** Thomas A. Richards.

**Supervision:** Thomas A. Richards.

**Validation:** David S. Milner, Jeremy G. Wideman.

**Visualization:** Jeremy G. Wideman.

**Writing – original draft:** David S. Milner, Jeremy G. Wideman, Thomas A. Richards.

**Writing – review & editing:** David S. Milner, Jeremy G. Wideman, Courtney W. Stairs, Cory D. Dunn, Thomas A. Richards.

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
