## [Editor Report · Decision Letter 0]

27 Jan 2021

Dear Dr Richards, 

Thank you for submitting your manuscript entitled "A functional mitochondrial-encoded restriction modification system in a heterotrophic protist" for consideration as a Research Article by PLOS Biology.

Your manuscript has now been evaluated by the PLOS Biology editorial staff, as well as by an academic editor with relevant expertise, and I'm writing to let you know that we would like to send your submission out for external peer review.

IMPORTANT: We will only consider your paper as a "Discovery Report" (you can find out more about this new-ish article type here: https://journals.plos.org/plosbiology/s/what-we-publish#loc-discovery-report). The rationale here is that I was initially considering rejecting your paper on the basis that the story is tantalising, but we could only consider it if you could demonstrate (rather than speculate on, however plausibly) the fact that this system can drive uniparental inheritance. It then occurred to me that this might work in its current form as a Discovery Report, with evidence for effects on mtDNA inheritance to potentially follow in a later "Update Article." I discussed this with the Academic Editor, who suggested that the latter could possibly be demonstrated heterologously in yeast, given the likely challenges of eliciting sexual reproduction in the protists...

Anyway, please could you change the article type to "Discovery Report" when you upload the additional metadata (see next paragraph).

Please re-submit your manuscript within two working days, i.e. by Jan 29 2021 11:59PM.

Kind regards,

Roli Roberts

Senior Editor

PLOS Biology

---

## [Decision Letter · Decision Letter 1]

3 Mar 2021

Dear Dr Richards,

Thank you very much for submitting your manuscript "A functional bacterial-derived restriction modification system in the mitochondrion of a heterotrophic protist" for consideration as a Discovery Report by PLOS Biology. As with all papers reviewed by the journal, yours was evaluated by the PLOS Biology editors as well as by an Academic Editor with relevant expertise and by four independent reviewers.

You'll see that all of the reviewers were intrigued by your findings and highly complimentary about the manuscript in general, but each has a number of requests that should be addressed before further consideration. Based on the reviews, we will probably accept this manuscript for publication, provided you satisfactorily address the remaining points raised by the reviewers. Please also make sure to address the following data and other policy-related requests.

IMPORTANT:

a) Please attend to the requests from the four reviewers.

b) We note that you say that you did not receive any specific funding for this work; can you confirm that this is the case?

c) Please address my Data Policy requests further down, i.e. provide the data that underlie the main and supplementary Figures, and cite the location of those data clearly in the respective legends.

We expect to receive your revised manuscript within two weeks, but would understand if it took a little longer in the current challenging circumstances. 

*Published Peer Review History*

*Early Version*

Sincerely,

Roli Roberts

Senior Editor,

rroberts@plos.org,

PLOS Biology

DATA POLICY:

Regardless of the method selected, please ensure that you provide the individual numerical values that underlie the summary data displayed in the following figure panels as they are essential for readers to assess your analysis and to reproduce it: Figs 2A (alignments), 3BCD, S1 (alignments), S2, S3, S4AB. NOTE: the numerical data provided should include all replicates AND the way in which the plotted mean and errors were derived (it should not present only the mean/average values).

DATA NOT SHOWN?

- Please note that per journal policy, we do not allow the mention of "data not sown", "personal communication", "manuscript in preparation" or other references to data that is not publicly available or contained within this manuscript. Please either remove mention of these data or add figures presenting the results and the data underlying the figure(s).

REVIEWERS' COMMENTS:

Reviewer #1:

This manuscript describes the presence of foreign genes in the mitochondrial genome a katablepharid protist, including an apparently functional restriction-modification system of bacterial origin. This represents a very unusual event in eukaryotic history that may provide insight into the acquisition and spread of selfish elements in mitochondrial genomes. Therefore, it has the potential to be of broad interest. I do not have experience with the functional characterization of restriction modification systems, but to my inexperienced eye, the transgenic work appeared to provide compelling evidence that the HpaII-like element and corresponding methyltransferase have retained their ancestral enzymatic activity inferred from sequence homology. Therefore, it is likely that they do constitute an intact restriction modification system in the protist. While I found the manuscript to be clearly presented and interesting, I do have some comments and concerns, which are listed below.

1. I agree with the authors' framing of the RM system as a likely selfish element that would be capable of preferentially spreading in the context of heteroplasmy with other mitochondrial haplotypes. However, I do not see the rationale for the authors' speculation that such a system was co-opted in the evolution of uniparental inheritance (last sentence of abstract). Yes, an RM system would result in biased outcomes when biparental inheritance and mitochondrial fusion are occurring. But I do not see that being the same thing as uniparental inheritance, which to me implies that there is a regulated system resulting in the consistent transmission of the mtDNA from just one sex, mating type, etc. For example, note that if an RM system was successful in spreading to fixation in a population with biparental inheritance, there would no longer be any bias in inheritance during sexual reproduction. If anything, much of the relevant literature has focused on the hypothesis that uniparental inheritance has evolved as a response to suppress such selfish elements. For these reasons, I would suggest revising the last sentence of the abstract and the paragraph starting on line 296 to focus on these as selfish genetic elements rather than a mechanism of uniparental inheritance.

2. Line 57-58. "no gain-of-function transfers into the mitochondrial genome have, to our knowledge, been previously reported." Perhaps I am misinterpreting the authors' meaning here, but I do not believe this is an accurate statement. What about the acquisition of mutS gene in the mitochondrial genomes of octocorals? Or the ORFs of likely viral origin in bivalves with doubly uniparental inheritance? There may be other examples, as this is not a topic I've ever tried to search out comprehensively in the literature or published mitogenomes. There are also other cases that may be relevant, though perhaps not directly contradictory to this statement. For example, there are many mitochondrial plasmids in plants, fungi, and ciliates that harbor a whole suite of foreign genes, many of which have been incorporated into mitochondrial chromosomes. Plant mitogenomes have also acquired many functional tRNAs from plastids and bacteria (perhaps some of the "replacement genes" that authors referred to). There is also at least one case where a nuclear-derived gene was shown to be transcribed in plant mitochondria (Qiu et al. 2014: https://www.sciencedirect.com/science/article/pii/S2214662814000073).

3. Line 189. "Domestication" is a loaded term with a lot of implications. The fact that codon usage is evolving toward a higher degree of similarity with the mitogenome could reflect something as simple as it being exposed to the same mutation pressures because the RM genes are now located in that genome. I would suggest revising this sentence to avoid suggesting that the changes in codon usage are evidence of domestication.

4. Katablepharid protists are a lineage that will be unfamiliar to most readers (including me). A little more content in the introduction on what they are and where they fall within the tree of life might be helpful.

5. Whereas other figures indicate that error bars represent one StdDev, this information in not provided in Fig S4. In addition to adding that information to the figure legend, I suggest more generally clarifying the extent to which replicates in these analyses are (non)independent. For example, are they performed with independent transformants or splitting of a single transformed line?

Reviewer #2:

This very nice manuscript by Milner and colleagues describes the horizontal transfer of a bacterial restriction modification system into the mitochondrial genome of a protist. The authors perform functional experiments to demonstrate that the transferred genes are functional and propose a hypothesis for how the acquisition of this restriction modification system may be behaving as a selfish genetic element.

The work is quite novel and beautifully done. The manuscript is clearly written and will be of great interest to the audience of PLoS Biology and of evolutionary biologists in general. I only have a few minor comments for the authors to consider:

- lines 36-38: what do the transferred genes have to do with uniparental inheritance is not at all clear from the abstract. Part of what's missing is a clear statement that the restriction modification system is a type of selfish element - the other part is explaining more the connection between selfishness and uniparental inheritance. The authors have explained these connections in the main text but not in the abstract.

- line 59" "Truly 'selfish' elements". What do you mean by "truly"? Are there selfish elements that are not truly selfish? and why the quotation marks for such a standard term?

- paragraph starting on line 111: in this paragraph, i would have appreciated some additional info on the taxonomic provenance of the taxa mentioned (e.g., Algibacter belongs to Flavobacteria). Whether the reported identity scores are at the amino acid or nucleotide level also needs to be reported

- line 148: "strong nucleotide identity" -> "high nucleotide identity"

- line 165: "limited bootstrap support" -> "low bootstrap support"

- line 176: "relatively extended branch" -> "long / longer branch"

- lines 177-8: i get how positive or relaxed selection could result in a long branch, but I did not understand how population bottlenecking could lead to (such) a long branch

- line 78, 262, and possibly elsewhere: Not sure what the policy of PLoS Biology is, but i generally find claims of priority ("the first known instance detected within a eukaryotic genome") not particularly helpful. I leave it up to the authors, but my advice would be to remove them from the manuscript and instead describe what their novel finding is and its significance.

Reviewer #3:

This is a very interesting paper presenting compelling evidence for there being a restriction modification system (which are usually found in bacteria) in the mitochondria of a eukaryote (specifically, a katablepharid protist). I only have a few minor comments:

Abstract: 

line 33: the connection to the flavobacteria is not overly compelling, as the phylogeny shows a long branch that connects pretty basally in the phylogeny, and I would suggest rewording to more accurately report the observation -- that phylogenetic analysis with known databases puts the closest relatives of the genes in the flavobacteria.

lines 37-8: there is very little in the paper about the possible co-option of RM systems by the nucleus to drive uniparental inheritance, and either this sentence should be deleted or modified, or the ideas fleshed out in the discussion. As it is, there is not much evidence presented for this (the Chlamydomonas work being ambiguous), and it seems a shame to end the abstract on such a dubious note, rather than focussing on the novel class of mitochondrial selfish elements.

line 174: What is meant by 'recent'? Can you rule out 100million years? 500 million? And can you rule out the possibility that there were 1 or more intermediary species between the flavobacterium ancestor and your protist that are not yet in the databases?

lines 268-271: be more explicit about the degeneration seen.

lines 297-300: This sentence does not make sense to me. Differential parental inheritance will occur while the RM system is rare in the population, but then when it is common and everyone has it, then there is no more differential inheritance -- it is a purely transient thing. Given that, it is not clear how important it is?

Reviewer #4:

Review of Milner et al. Plos Biology 2021

The paper describes a case of horizontal gene transfer of a two-gene module the authors later go on to confirm is involved in methylation and degradation of DNA with precise motifs. Authors infer that this HGT results in mitochondrial drive.

This is an exceptionally well-written manuscript, which takes a very simple observation of anomalous sequence in a genome and successfully tests a number inferences about the significance of its existence. The work is boosted by tests at multiple scales, e.g. expression of the sequence in global metagenomes, heterologous expression/characterization of the sequence and its pre-HGT bacterial counterpart, imaging of co-location of target DNA and enzyme expression, codon evolution, etc. There are a number of broadly interesting outcomes, including exceptional events in molecular evolution (expansion of organelle genome), cooperation between selfish genetic elements and host, and a novel mechanism of the origin of genetic drive.

I only have a couple of comments. 

"SAG" was not intuitive in the figure without reading multiple points of the text. Figure 1 especially would be much more readily grasped with an indication what SAG means/implies. Either spelling out the phrase, or adding a modifier to the mitochondrial genome names that indicates what is being distinguished by the re-sequencings.

Along with this, there is room for misinterpretation in line 140-141. The sentence is describing the reproducibility, but could be interpreted to mean independent evolutionary events. I think this would be avoided by stating the result more plainly and without combining with the inference. Something more like "amplification of the bacteria-like RM systems from mitochondrial genomes was independently replicated 5 times."

The legend in Figure 2A suggests Bayesian posterior probabilities can exceed 1 ("black dot > 1/90"). This should be more clear.

The "domestication" referred to in line 189 has the specific term "amelioration" assigned to it. I am not sure which term brings more unnecessary implication, but the latter would be better connected to the appropriate literature.

---

## [Editor Report · Decision Letter 2]

23 Mar 2021

Dear Dr Richards,

On behalf of my colleagues and the Academic Editor, Harmit Malik, I'm pleased to say that we can in principle offer to publish your Research Article "A functional bacteria-derived restriction modification system in the mitochondrion of a heterotrophic protist" in PLOS Biology, provided you address any remaining formatting and reporting issues. These will be detailed in an email that will follow this letter and that you will usually receive within 2-3 business days, during which time no action is required from you. Please note that we will not be able to formally accept your manuscript and schedule it for publication until you have made the required changes.

PRESS: We frequently collaborate with press offices. If your institution or institutions have a press office, please notify them about your upcoming paper at this point, to enable them to help maximise its impact. If the press office is planning to promote your findings, we would be grateful if they could coordinate with biologypress@plos.org. If you have not yet opted out of the early version process, we ask that you notify us immediately of any press plans so that we may do so on your behalf.

Thank you again for supporting Open Access publishing. We look forward to publishing your paper in PLOS Biology. 

Sincerely, 

Roli Roberts

Roland G Roberts, PhD 

Senior Editor 

PLOS Biology